Occasional hybridization between a native and invasive Senecio species in Australia is unlikely to contribute to invasive success

Dormontt Eleanor E. eleanor.dormontt@adelaide.edu.au 1
Prentis Peter J. 2
Gardner Michael G. 3
Lowe Andrew J. andrew.lowe@adelaide.edu.au 1
1 The Environment Institute, School of Biological Sciences, University of Adelaide , Adelaide , South Australia , Australia
2 Institute for Future Environments, School of Earth, Environmental and Biological Sciences, Queensland University of Technology , Brisbane , Queensland , Australia
3 School of Biological Sciences, Flinders University of South Australia , Adelaide , South Australia , Australia
Daehler Curtis
Electronic publication date: 2017 Aug 15
Publication date: 2017
Volume: 5
Electronic Location ID: e3630
Received 2017 Feb 15; Accepted 2017 Jul 11
Copyright: ©2017 Dormontt et al.
Copyright year: 2017
Copyright holder: Dormontt et al.
License: This is an open access article distributed under the terms of the Creative Commons Attribution License, which permits unrestricted use, distribution, reproduction and adaptation in any medium and for any purpose provided that it is properly attributed. For attribution, the original author(s), title, publication source (PeerJ) and either DOI or URL of the article must be cited.
License URL: https://creativecommons.org/licenses/by/4.0/

Keywords: Introgression, Biological invasions, AFLP, Microsatellites, Fireweed

Funding: Australian Research Council DP0664967 Funding for this work was provided by the Australian Research Council (grant number DP0664967) awarded to AJL. The funders had no role in study design, data collection and analysis, decision to publish, or preparation of the manuscript.

==============================
Background

Hybridization between native and invasive species can facilitate introgression of native genes that increase invasive potential by providing exotic species with pre-adapted genes suitable for new environments. In this study we assessed the outcome of hybridization between native Senecio pinnatifolius var. pinnatifolius A.Rich. (dune ecotype) and invasive Senecio madagascariensis Poir. to investigate the potential for introgression of adaptive genes to have facilitated S. madagascariensis spread in Australia.

Methods

We used amplified fragment length polymorphisms (141 loci) and nuclear microsatellites (2 loci) to genotype a total of 118 adults and 223 seeds from S. pinnatifolius var.pinnatifolius and S. madagascariensis at one allopatric and two shared sites. We used model based clustering and assignment methods to establish whether hybrid seed set and mature hybrids occur in the field.

Results

We detected no adult hybrids in any population. Low incidence of hybrid seed set was found at Lennox Head where the contact zone overlapped for 20 m (6% and 22% of total seeds sampled for S. pinnatifolius var. pinnatifolius and S. madagascariensis respectively). One hybrid seed was detected at Ballina where a gap of approximately 150 m was present between species (2% of total seeds sampled for S. madagascariensis).

Conclusions

We found no evidence of adult hybrid plants at two shared sites. Hybrid seed set from both species was identified at low levels. Based on these findings we conclude that introgression of adaptive genes from S. pinnatifolius var. pinnatifolius is unlikely to have facilitated S. madagascariensis invasions in Australia. Revisitation of one site after two years could find no remaining S. pinnatifolius var. pinnatifolius, suggesting that contact zones between these species are dynamic and that S. pinnatifolius var. pinnatifolius may be at risk of displacement by S. madagascariensis in coastal areas.

Introduction

The study of hybridization between related species has continued to fascinate biologists since the early 19th century (Stebbins, 1959) with the potential role of hybridization in evolutionary diversification of particular interest (Abbott et al., 2013; Anderson & Stebbins Jr, 1954; Arnold, 2004; Seehausen, 2004; Yakimowski & Rieseberg, 2014). Hybridization can have diverse outcomes including the formation or extinction of species (Abbott et al., 2010; Rhymer & Simberloff, 1996; Todesco et al., 2016), introgression of genes from one parental taxa to another (e.g., Whitney, Randell & Rieseberg, 2006; Whitney, Randell & Rieseberg, 2010), and demographic swamping (e.g., Field et al., 2008; Prentis et al., 2007). Alternatively, successful hybridization between co-occurring species may be rare enough to have little long-term impact on either parental taxa.

Hybridization between native and invasive species is of particular interest, indeed in their seminal review, Ellstrand & Schierenbeck (2000) argue that hybridization (inter- and intra-specific) can act as a stimulus for the evolution of invasiveness. One mechanism by which this can occur is through introgression of adaptive genes resulting from hybridization followed by repeated backcrossing with parental taxa. Introgression of native genes can increase invasive potential by providing exotic species with pre-adapted genes suitable for new environments (e.g., Whitney, Randell & Rieseberg, 2010), conversely introgression of exotic genes can facilitate the transfer of weedy traits to native species, jeopardizing genetic integrity (e.g., Fitzpatrick et al., 2010). Simulation studies on neutral genes have revealed that the very nature of the invasive process is likely to promote almost exclusively unidirectional introgression, from the native species into the invader (Currat et al., 2008) increasing the likelihood of locally adapted genes facilitating invasive species spread. Aside from introgression, hybrid progeny can go on to become invasive species in their own right, such as Senecio squalidus which evolved via homoploid hybrid speciation from the parental species Senecio aethnensis and Senecio chrysanthemifolius (Abbott et al., 2010). In extreme cases, hybrid progeny can be so successful that they completely displace their parental species in the field, such as the Californian wild radish, an invasive hybrid lineage derived from introduced Raphanus sativus and Raphanus raphanistrum (Hegde et al., 2006).

In the current study, we focus on a native and invasive species pair, Senecio pinnatifolius  var. pinnatifolius A. Rich. (dune ecotype) and Senecio madagascariensis Poir., which co-occur along ∼2,000 km of coast line in New South Wales, Australia. Senecio madagascariensis is a successful invasive plant in Australia and typically a weed of agricultural pastures, however it can also be found growing alongside the native S. pinnatifolius in natural systems, raising the possibility that introgression of adaptive genes from the native has facilitated its spread into these areas. In previous work on S. pinnatifolius var. serratus (tableland ecotype) Prentis et al. (2007) found hybrid seed set but no adult hybrids in the field. Prentis et al. (2007) also modelled loss of viable seeds to hybridization and predicted the eventual displacement of S. pinnatifolius var. serratus by the invasive S. madagascariensis at their study sites.

Whether Prentis et al.’s (2007) conclusions are more broadly applicable to other S. pinnatifolius ecotypes is not clear. Reports of potential hybrids between S. madagascariensis and S. pinnatifolius var. pinnatifolius (dune ecotype) (Scott, 1994; EM White, Queensland University of Technology, Australia, pers. comm., 2005) have served as a stimulus for the current study which sought to assess the incidence of hybridization between S. madagascariensis and S. pinnatifolius var. pinnatifolius at two sites where the species co-occur. The species have overlapping flower times (Radford, 1997; Radford & Cousens, 2000) and share pollinators (White, 2008) making hybridization in the field possible. The two species do possess different ploidy however (S. pinnatifolius is tetraploid, S. madagascariensis diploid) with the triploid hybrid offspring that would most often result, typically having low fertility due to meiotic pairing problems between homeologous chromosomes. Despite the low fertility typical of triploid hybrids, they have been shown to act as a bridge between taxa facilitating introgression (Koutecký et al., 2011; Lowe & Abbott, 2000) and it may be possible for S. madagascariensis to produce unreduced gametes (Brownfield & Köhler, 2010; Koutecký et al., 2011; Ramsey & Schemske, 1998) that could fuse with normal S. pinnatifolius gametes to produce tetraploid hybrid offspring.

As a successful invader, S. madagascariensis is spreading through south-eastern Australia into new habitats, possibly due in part to introgression of adaptive genes from S. pinnatifolius var. pinnatifolius. As a first step to examining this possibility, we assessed the extent of hybrid seed set and the incidence of adult hybrids at our field sites. We hypothesized that if introgression was occurring between S. pinnatifolius var. pinnatifolius and S. madagascariensis, then evidence of mature hybrids should exists at sites where the two species co-occur.

We use amplified fragment length polymorphisms (AFLPs) and microsatellites from mature individuals of both species at two sites where the species co-occur and one site for each species that was at least 2 km away from any other known populations of the congener (‘allopatric’). We also sampled open pollinated progeny arrays in areas of co-occurrence. We asked whether hybrid seed set occurs in the field and whether adult hybrids are present that could backcross with either species to facilitate introgression.

Materials & Methods

Study species

Senecio madagascariensis is a diploid plant, initially introduced to south-eastern Australia from the KwaZulu-Natal province of South Africa in the early part of the 20th century (Radford et al., 2000). Molecular analysis of contemporary and historical field collections has pointed to at least two separate introductions (Dormontt et al., 2014). Senecio pinnatifolius (previously S. lautus) is a tetraploid plant native to Australia. There have been multiple taxonomic treatments of the species complex (Ali, 1969; Radford, 1997; Roda et al., 2013; Thompson, 2005), with each agreeing on distinction of ‘dune’, ‘headland’ and ‘tableland ecotypes. Senecio pinnatifolius var. pinnatifolius (dune ecotype) occurs on coastal sands along the east coast of Australia and is the only S. pinnatifolius ecotype analysed in the current study, hereafter referred to only as Senecio pinnatifolius. Initially included in the S. pinnatifolius complex, S. madagascariensis was recognised as a separate species after Hilliard’s (1977) treatment of Asteraceae in Natal (Sindel et al., 1998). This separation has been supported by morphological comparisons (Thompson, 2005) and cytological studies (Radford, Liu & Michael, 1995) finding 2n = 20 for S. madagascariensis and 2n = 40 for S. pinnatifolius.

Both species are considered annuals or short lived perennials (Radford & Cousens, 2000), look superficially identical, growing to approximately 0.6 m and with bright green leaves and yellow inflorescences that are heterogamous and radiate. The species can be reliably distinguished by the number of involucral bracts present, 18–21 in S. madagascariensis, 11–14 in S. pinnatifolius. Both species are outcrossing, self-incompatible (Ali, 1966) and insect pollinated, predominantly by the introduced European honey bee Apis mellifera and various species of Syrphidae (White, 2008); seeds are wind dispersed. In comparisons of life history traits between the species, S. madagascariensis was found to perform better than S. pinnatifolius with respect to seedling, growth and fecundity measures but S pinnatifolius maintained a stronger soil seed bank (Radford & Cousens, 2000).

Sample collection and seed germination

One allopatric population of each species was sampled along with two sites where the two species occurred together (Table 1). At each site, twenty individuals of each species were sampled with fresh leaf material preserved in silica beads for DNA extraction. Additionally, at shared sites, multiple mature seed heads were collected from sampled plants where available and stored for later germination. Plants were sampled in a systematic fashion (across the contact zone at shared sites) and the location of all plants recorded with GPS (with the exception of the allopatric S. madagascariensis population where coordinates were not recorded). Plants that were not sampled for DNA were identified in the field and GPS coordinates recorded. The GPS recorded relative position to within 1 m accuracy. Two years after initial sampling, we revisited one of the shared sites to survey the changes in abundance of S. pinnatifolius and S. madagascariensis.

Table 1 Information on sites and samples included in the study.

Number of adults sampled (na), number of adults with genotyped seedlings (ns), number of seedlings analysed (s) and range of seedlings genotyped per mother plant.

Species	Population	Latitude	Longitude	na	ns	s (range)	
S. pinnatifolius	Southport	S27°56′15″	E153°25′35″	20	0	–	
	Lennox Head	S28°47′9″	E153°35′38″	20	7	52 (2–10)	
	Ballina	S28°52′25″	E153°35′21″	20	7	61 (4–10)	
S. madagascariensis	Oxenford	S27° 53′23″	E153°18′43″	18	0	−	
	Lennox Head	S28°47′9″	E153°35′38″	20	6	59 (10–19)	
	Ballina	S28°52′25″	E153°35′21″	19	6	51 (4–10)	
			Total	117	26	223 (2–19)	

In the laboratory, mature achenes were detached from their pappus and the seed coat nicked with a scalpel. Seeds were grown on moist filter paper with gibberellic acid (GA3) in a 12 h photoperiod at 25 °C to stimulate germination. All germinated seedlings, up to a maximum of ten per parent plant, were frozen at −80 °C prior to DNA extraction, except in one case where 20 seedlings were used (Table 1, see results for further explanation). The number of seeds per parent plant that successfully germinated were classified as ‘ten or more’, or ‘less than ten’. Germination rates were compared between sites using a permutation approach in Resampling Stats Add-In for Excel v4.0 (https://www.statistics.com/). Seeds per parent plant that successfully germinated (‘ten or more’, or ‘less than ten’) were sampled without replacement to simulate the same number of parent plants per site as the empirical data. The proportion of ‘less than ten’ seedlings per site were compared in 10,000 simulations to the empirical data to obtain estimated P values.

Table 2 Details of final round PCR primers used in study.

Markers used were amplified fragment length polymorphisms (AFLP), nuclear microsatellites (nSSR), and one chloroplast microsatellite (cpSSR). Primer information includes primer type (EcoRI or Mse origin for AFLPs, locus name and primer direction for microsatellites); primer sequence including fluorescent dye (PET, FAM, NED or VIC); annealing temperature (Ta); and number of loci (for AFLPs) or alleles (for microsatellites) scored for each pair (n).

Marker	Primer #1	Primer #2	Ta (°C)	n	
AFLP	EcoRI	TACTGCGTACCAATTCAGC(PET)	Mse	GACGATGAGTCCTGAGTAACAA	65–56	48	
	EcoRI	TACTGCGTACCAATTCAGC(FAM)	Mse	GACGATGAGTCCTGAGTAACAG	65–56	57	
	EcoRI	TACTGCGTACCAATTCAGC(NED)	Mse	GACGATGAGTCCTGAGTAACCG	65–56	37	
nSSR	Se-116F	CCTTCTGGTTGATTTGGCTAAGC(FAM)	Se-116R	AGAACTGCACATTTGAAGCCTG	48	15	
	Se-138F	ACTTCGTGGGCCATTCCAG(VIC)	Se-138R	CTTCCTGCATAACATCCACCAC	58	24	
cpSSR	Ccmp3F	CAGACCAAAAGCTGACATAG(PET)	Ccmp3R	GTTTCATTCGGCTCCTTTAT	50	3	

Genetic analysis

DNA extractions were carried out using the Machery-Nagel Nucleospin Plant II Kit with the PL2/PL3 buffer system. Two published microsatellite loci (Le Roux & Wieczorek, 2007) originally developed for S. madagascariensis and found to be cross compatible with S. pinnatifolius, were used to screen all adults and seedlings from both species (Table 2). PCR reactions were prepared with ∼20 ng of template DNA, 1× reaction buffer, 0.2 mM of each dNTP, 2.5 mM MgCl2, 0.4 µM of each primer, and 0.02 U Amplitaq Gold® (Applied Biosystems, Foster City, CA, USA) to give a final PCR reaction volume of 10 µL. Reactions involved an initial denaturation step of 94  °C for 2 min, followed by 35 cycles at 94 °C for 1 min, the loci specific annealing temperature for 1 min (Table 2), 72 °C for 1 min and 30 s, and a final extension at 72 °C for 30 min. One published chloroplast microsatellite locus (Weising & Gardner, 1999) (Table 2) was found to produce bands mutually exclusive to S. pinnatifolius and S. madagascariensis and so was included to allow identification of the maternal parent of any hybrid adults detected in the field. Reactions were prepared with ∼20 ng of template DNA, 1x reaction buffer, 0.2 mM of each dNTP, 2.5 mM MgCl2, 0.5 µM of each primer, and 1 U IMMOLASE™ DNA polymerase (Bioline, London, UK) to give a final PCR reaction volume of 10 µL. Reactions involved an initial denaturation step of 94 °C for 5 min, 30 cycles of 94 °C for 20 s, 50 °C for 20 s, 72 °C for 20 seconds, and a final extension at 72 °C for 30 min. Products were separated using the ABI 3730 DNA analyzer (Applied Biosystems, Foster City, CA, USA) with the GeneScan™—500 LIZ® size standard. Genemapper® Software v4.0 (Applied Biosystems, Foster City, CA, USA) was used to score fragments. Scoring was recorded in a binary matrix with presence or absence of particular alleles indicated by a 1 or 0. This method allowed for polyploidy and diploid data to be directly compared and analysed together. DNA from thirty one individuals (9% of samples) were amplified twice for microsatellite analysis to enable estimation of error rates, calculated according to DeWoody, Nason & Hipkins (2006).

Amplified fragment length polymorphisms (AFLPs) were assessed according to the method of Vos et al. (1995) with modifications. Restriction digests were performed in 20 µl reactions with ∼200 ng of DNA, 1× restriction digest buffer 2, 10 U MseI (New England Biolabs, Ipswich, MA, USA), 10 U EcoRI (New England Biolabs, Ipswich, MA, USA), and 1×BSA. Reactions were incubated for 3 h at 37 °C, followed by 20 min at 65 °C to denature the enzymes. Adapters were ligated to the digested fragments in reactions containing 20 µl of digested DNA, 1×T4 ligase buffer, 2.5 µM EcoRI adapter, 0.25 µM MseI adapter and 3 U of T4 DNA ligase (New England Biolabs, Ipswich, MA, USA). Reactions were incubated overnight at 16 °C.

Pre-selective amplifications contained 2 µl of digested and ligated DNA, 1×Optimised DyNAzyme™ EXT buffer (including 1.5 mM Mg2+), 0.2 mM of each dNTP, 0.5 μM MseI (+C), 0.5 μM EcoRI (+A) primers and 0.25 U DyNAzyme™ EXT DNA polymerase to give a final PCR reaction volume of 25 µL. Reactions involved an initial denaturation step of 75 °C for 2 min, then 20 cycles of 94 °C for 30 s, 56  °C for 30 s, 75 °C for 2 min, and a final extension at 60 °C for 30 min. PCR products were run on agarose gel to check for successful amplification.

Selective amplifications contained 1 µl of 1 in 30 diluted pre-selective PCR product, 1×TaqGold buffer (Applied Biosystems, Foster City, CA, USA), 2 mM MgCl2, 0.2 mM of each dNTP, 0.3 µM MseI + 3bp primers, 0.3 µM EcoRI +3 bp primers and 0.75 U TaqGold (Applied Biosystems, Foster City, CA, USA) in a final PCR reaction volume of 15 µL. Reactions involved an initial denaturation step of 94 °C for 2 min, then 10 cycles of 94 °C for 30 s, 65−56 °C for 30 s (reduce by 1  °C per cycle), 72 °C for 2 min, then 26 cycles of 94 °C for 30 s, 56 °C for 30 s, 72 °C for 2 min and a final extension at 60°C for 5 min. Twelve selective amplifications were trialled using a range of +3 bp primer combinations on four individuals of each species. Products were run on 5% acrylamide gels using a Gelscan GS2000 (Corbet Research) and the three most suitable combinations (based on appropriate number and strength of bands, polymorphisms and ease of scoring) were chosen for selective amplification of all samples (Table 2). Products were separated using the ABI 3730 DNA analyzer (Applied Biosystems) with the GeneScan™—500 LIZ® size standard. Forty one adult individuals (12% of total individuals) were re-extracted for DNA and the AFLP process repeated to allow loci validation and error rate calculations. Vegetative material from seedlings was too small to allow for repeated extractions, so only adults were used. A negative control was included throughout the extraction/AFLP process to enable exclusion of non-specific bands.

Genemapper® Software v4.0 (Applied Biosystems, Foster City, CA, USA) was used to manually allocate bins to appropriate loci, all duplicated samples were visualised and where consistent banding was apparent between samples, this was assigned as a specific locus. Once manual binning was complete, the full dataset was automatically scored using Genemapper® Software v4.0 (Applied Biosystems, Foster City, CA, USA) and raw peak height data obtained. The raw peak height data were then used with AFLPScore v1.4 (Whitlock et al., 2008) to minimise error whilst maximising number of retained loci. AFLPScore allows the user to select a range of loci selection thresholds (the average intensity of bands at a specific locus, above which a locus is retained in the dataset) and phenotype calling thresholds (the intensity of a given band, either in absolute terms, or as a percentage of the average for that locus, above which band presence will be called). By comparing combinations of different locus selection and phenotype calling thresholds, the user can select thresholds which result in reduced error and maximised retained loci. After error reduction via AFLPscore, a phenotype matrix was exported and loci with the highest error rates systematically removed to create 11 separate datasets with error rates of 0%, 1%, 2%, 3% etc. up to 10% and additionally one with a 17% error rate (the output from AFLPscore with no loci removed). To assess the effects of each error rate on overall information content, the data from the allopatric populations of each species were analysed using the program STRUCTURE (Pritchard, Stephens & Donnelly, 2000) with RECESSIVEALLELES set to 1 to account for dominant data (Falush, Stephens & Pritchard, 2007). Number of predefined populations (K) was set from 1 to 5. Each run consisted of a burn-in period of 100,000 Markov Chain Monte Carlo (MCMC) repetitions, followed by 1,000,000 MCMC repetitions, the program was run five times to allow averaging of results in CLUMPP (Jakobsson & Rosenberg, 2007). Plots were displayed in DISTRUCT (Rosenberg, 2004). The final dataset was chosen based on how well it could detect the expected structure (designation of K = 2, highest probability of individuals belonging to the appropriate species cluster) and how robust it was to the negative impacts of higher error (such as the signal from plate effects) see Zhang & Hare (2012) for an in-depth discussion and analysis of this approach.

Data analysis

To assess hybridisation, the AFLP and microsatellite data were combined into one data matrix, in the case of the microsatellites, each allele was either designated as present or absent. Assignment of an individual as either a pure parental species or a hybrid was based on a consensus between two different analysis methods, with the most conservative (i.e., non-hybrid) designation accepted if results were inconsistent between methods. The first method used the allocation procedure in the program AFLPOP (Duchesne & Bernatchez, 2002). The allopatric populations of each species were set as sources, and the remaining samples allocated to either one of the pure species or hybrid origin by the program. Zero frequencies were corrected as 1∕n + 1, where n is the sample size. The allocation minimal log-likelihood difference (MLD) was initially set to 1 (meaning allocation only occurred when designation was 10 times more likely than any other possible origin). Samples that could not be allocated in this way were re-run with MLD set to 0 (allocating to highest likelihood source regardless of the magnitude of difference between alternate likelihoods).

The second method used the program STRUCTURE (Pritchard, Stephens & Donnelly, 2000) with extensions implemented by Falush, Stephens & Pritchard (2007) to account for genotypic ambiguity that is inherent in dominant markers; RECESSIVEALLELES was set to 1. STRUCTURE has been used successfully to assess datasets comprised of individuals with different ploidy levels (De Hert et al., 2012; Pinheiro et al., 2010; Zalapa et al., 2011). Number of predefined populations (K) was set to 2. Each run consisted of a burn-in period of 100,000 Markov Chain Monte Carlo (MCMC) repetitions, followed by 1,000,000 MCMC repetitions, the program was run five times to allow averaging of results in CLUMPP (Jakobsson & Rosenberg, 2007). Plots were displayed in DISTRUCT (Rosenberg, 2004). Clustering of adult and seedling genotypes of each species at both allopatric and shared sites were visualised with a principal coordinate analysis (PCoA) in GENALEX v6.4 (Peakall & Smouse, 2006; Peakall & Smouse, 2012). Hybrid zone mapping was completed using ArcGIS v9.2 (ESRI, Redlands, CA, USA).

Results

Loci selection

Both nuclear microsatellite loci were polymorphic in both species and retained for further analysis (Table 2). The single chloroplast microsatellite locus was polymorphic in S. pinnatifolius (two alleles) and monomorphic in S. madagascariensis but alleles were not shared between species. All adults and seedlings genotyped conformed to their expected species specific chloroplast haplotypes. The observed error rate per allele and per locus for the nuclear microsatellites was zero. Of the 12 AFLP primer combinations trialled, three were chosen for screening all samples (Table 2).

In AFLPScore v1.4 (Whitlock et al., 2008), mismatch error rates were used to optimise scoring parameters using both absolute and relative phenotype calling thresholds on an initial dataset containing 247 loci. The error rate of the exported data set was 0.17 with 233 retained loci (Data S1), achieved by filtering data using an absolute phenotype-calling threshold of 250 relative fluorescence units (RFU), prior to application of a 50 RFU locus-selection threshold. After STRUCTURE analysis, the data set equating to an average error rate of 6% was chosen, as it correctly identified K = 2, indicated high assignment rates of individuals to their correct species, did not display any significant plate effects at K = 3 (the number of plates) and contained a reasonable number of loci (142) (Fig. S1). An overall error rate of 6% is high compared to the 2–5% reported for most AFLP studies (Bonin et al., 2004) but under the maximum threshold of 10% recommended by Bonin, Ehrich & Manel (2007) (Fig. S2). Systematically evaluating the effects of different error rates on result and selecting that which is most informative and least confounding allows the information content of the dataset to be maximised without limiting the included loci in order to conform to an arbitrary cut off point (Zhang & Hare, 2012).

Hybridisation

No adult hybrids were detected in the field. In total, 17 hybrids were observed from 223 seeds (8% of seeds and 5% of all individuals genotyped including adults). Fourteen of these hybrid seeds were from a single S. madagascariensis mother and three from two S. pinnatifolius mothers (Fig. 1). Hybrid seed set was observed at Lennox Head where 6% and 22% of the total seeds sampled for each species at that site were hybrid for S. pinnatifolius and S. madagascariensis respectively. For each adult with a hybrid seed set, the distance to the nearest congeneric was <15 m (Fig. 1). There was uncertainty in the field about seeds collected from what appeared to be a single plant but may have been two adjacent plants. Twenty seeds were germinated from this sample with the hope that separation of individuals could be made in the lab from the results of the genetic analysis. The microsatellite data confirmed that these seeds did indeed come from the same individual, and so one adult has 20 genotyped offspring instead of the usual 10. A single hybrid seed was detected at Ballina from a S. madagascariensis mother (Fig. 1), as designated by agreement between AFLPop and STRUCTURE, however this individual does closely cluster with other pure S. madagascariensis seedlings in the PCoA analysis (Fig. 2), which may indicate a false positive result. AFLPOP (Duchesne & Bernatchez, 2002) allocated 90% of adults and 68% of seedlings with a minimal log-likelihood difference (MLD) of 1 (indicating that the allocation was at least 10 times more likely than any other). The remaining samples allocated with MLD set to 0. One S. madagascariensis seedling was allocated to S. pinnatifolius with MLD set to 0. The chloroplast haplotype of this individual was consistent with S. madagascariensis maternity and it clustered with the hybrid seedlings in the PCoA (Fig. 2), so has been designated as a hybrid. Hybrid origin was more conservatively allocated in the program STRUCTURE (Pritchard, Stephens & Donnelly, 2000) (Fig. 3) with 92% consensus between the two methods. Final designation used the most conservative (non-hybrid) allocation.

Figure 1 Location of samples at shared field sites.

(A) shows location of sites in Australia; (B) shows location of sites in New South Wales; (C) shows the Ballina site; (D) shows the Lennox Head site. Senecio pinnatifolius is depicted with white symbols, Senecio madagascariensis with black symbols. The position of un-sampled plants is shown by crosses, small circles are genotyped adult plants, and larger circles are genotyped plants with genotyped seed. The proportion of seeds with pure or hybrid origin is shown in the large circles, grey indicating hybrid. Where hybrids occur, call out boxes enlarge this detail. The number of seeds sampled per adult (n) is indicated.

Figure 2 Principal coordinates analysis.

Clustering of adults (A) and seeds (B) of Senecio pinnatifolius (circles) and Senecio madagascariensis (triangles) at allopatric (grey) and the two shared sites: Lennox Head (white) and Ballina (black). Hybrid designation is based on the combined results from STRUCTURE and AFLPop. Hybrid seeds were found at Lennox Head with S. pinnatifolius mothers (+) and S. madagascariensis mothers (X). One hybrid with an S.madagascariensis mother was found at Ballina ( ).

Figure 3 Data output from the program STRUCTURE, runs averaged with CLUMPP and displayed with DISTRUCT.

(A) shows data for Senecio pinnatifolius and (B) for Senecio madagascariensis. Locations, and whether the samples were adults or seeds, are shown under the bar plots. For both species, the allopatric population is shown first, followed by the shared sites. Individuals designated as hybrid in the final dataset are indicated with an asterisk.

Germination and site revisit

Germination success varied somewhat between individuals with 11–38% of parent plants per site producing less than ten seedlings for DNA extraction. However, the simulation approach used to examine germination rates found no significant differences between germination rates at each site and those expected to arise randomly. The Lennox Head site was revisited in 2009, two years after initial sampling and a morphological survey of plant species identity undertaken. All plants observed were identified as S. madagascariensis.

Discussion

Hybridisation between native and exotic species can affect biological invasions in several ways, including via introgression (Currat et al., 2008; Prentis et al., 2008; Whitney, Randell & Rieseberg, 2006; Whitney, Randell & Rieseberg, 2010) and pollen swamping (Buggs & Pannell, 2006; Petit et al., 2004; Prentis et al., 2007). Despite occasional hybrid seed set between native Senecio pinnatifolius var. pinnatifolius (dune ecotype) and invasive Senecio madagascariensis, we found no evidence to support the role of introgression in this system. We found very low levels of hybrid seed formation in both S. pinnatifolius and S. madagascariensis mothers at one site (Lennox Head, NSW) where the two species occur together with a minimum distance of approximately 15 m. A single S. madagascariensis mother and two S. pinnatifolius mothers produced all the hybrid seeds at this site. At the other study site (Ballina, NSW) a single hybrid seed was detected in a S. madagascariensis mother, with a distance of approximately 155 m to the closest S. pinnatifolius plant. The observed imbalances in hybrid seed set amongst conspecifics may be the result of proximity to congenerics (Fig. 1) or could indicate individual variation in ability to set hybrid seed.

No adult hybrids were identified at either site which could be explained simply by the low overall hybrid seed set observed; perhaps hybrid adults were not present simply by chance in the study year. Prentis et al. (2007) found no significant differences in viability between seeds generated from intra- and inter-specific crosses suggesting that any fitness costs are incurred after germination. However, Prentis et al. (2007) examined S. pinnatifolius var. serratus, not S. pinnatifolius var. pinnatifolius (the focus of this study), so it is possible that hybrid seed viability varies between S. pinnatifolius varieties. Reciprocal crossing experiments between S. madagascariensis and the different S. pinnatifolius varieties would further explore this issue. Alternatively, the lack of mature hybrids observed in the field could be the result of reduced hybrid fitness acting as a post-zygotic mating barrier between S. pinnatifolius and S. madagascariensis. Previous work has shown that synthetic hybrids between the two species grown under glasshouse conditions had low viability and were sterile (Radford, 1997). Occasional adult hybrid occurrence may explain the findings of Scott (1994) and EM White, Queensland University of Technology, Australia, pers. comm., 2005, who report observation of putative hybrid plants.

The present study did not identify the ploidy level of the hybrid seedlings identified from our open pollinated progeny arrays which would be an interesting topic for further research. Koutecký et al. (2011) found that hybrids formed from reduced gametes between diploid Centaurea pseudophrygia and tetraploid Centaurea jacea were less common in the seed set of maternal plants but more common in the adult hybrid plants found in the field, suggestive of increased fitness of the tetraploid hybrids. These tetraploids were also able to backcross with C. jacea, facilitating introgression of C. pseudophrygia genes into C. jacea. In S. pinnatifolius x S. madagascariensis hybrids there may be similar fitness asymmetries associated with ploidy level but as hybridisation rates were so low and no adult hybrids were detected in the present study, the impact of any such differences is likely minimal.

Selection against hybrids in the field would constitute a post-zygotic mating barrier, yet the prevailing view is that pre-zygotic mating barriers are stronger in flowering plants (Dell’Olivo et al., 2011; Rieseberg & Willis, 2007; Widmer, Lexer & Cozzolino, 2009). However, global change (including increased movement of exotic species) is predicted to increase opportunities for hybridization through the erosion of pre-zygotic barriers (Vallejo-Marín & Hiscock, 2016). Evidence for pre-zygotic isolation barriers between S. pinnatifolius and S. madagascariensis are sparse at present. The two species can be found occurring in shared sites (Prentis et al., 2007; Radford, 1997; White, 2008), their flowering times overlap ((Radford, 1997; Radford & Cousens, 2000) and a similar suit of pollinators visit both species (White, 2008). However, it should be noted that reproductive isolation can still be favoured even when flowering times overlap but not completely, as is the case with S. pinnatifolius and S. madagascariensis (Radford, 1997). Even when pollinators are shared, they may prefer conspecific over heterospecific visitation (White, 2008). The relative contribution of these potentially reproductively-isolating barriers remains to be tested in this system with more extensive field and laboratory studies, incorporating greater geographical and temporal breadth. The use of AFLP and microsatellite markers to explore hybridization as implemented in this study have also now been superseded by genomic techniques that utilize next generation sequencing (NGS) methods to develop datasets with orders of magnitude more information (Goulet, Roda & Hopkins, 2017; Payseur & Rieseberg, 2016). Further work on the system should exploit these resources to better characterise hybridisation outcomes.

We set out to explore whether hybridisation between S. pinnatifolius var. pinnatifolius (dune ecotype) and S. madagascariensis was likely to have facilitated the spread of S. madagascariensis by way of introgression of adaptive genes. Due to the very low level of hybrid seed set and the absence of adult hybrids, we must conclude that introgression via fertile hybrids in the field is probably rare, at least at the sites we studied. As only two field sites were included it is difficult to generalise across the entire ∼2,000 km range in which the two species overlap, however we can tentatively support the findings of Prentis et al. (2007) who found similar results in their study of hybridisation between S. pinnatifolius var. serratus (tableland ecotype) and S. madagascariensis. It may be the case that all S. pinnatifolius ecotypes exhibit the same patterns when in sympatry with S. madagascariensis.

In their modelling of these hybrid zones Prentis et al. (2007) also predicted a demographic swamping of S. pinnatifolius by S. madagascariensis assuming that hybridisation is plant density dependent. However, the very low levels of hybrid seed set observed in S. pinnatifolius in the current study (6% of seeds) are not consistent with this prediction. The assumption of density dependence could not be verified as overall levels of hybridisation were too low but we did find the greatest proportion of hybrid seed set in an area of high congeneric plant density (Fig. 1). To adequately assess the density dependent nature of hybridisation, artificial manipulation of plant densities in open pollinated conditions would be required. At our subsequent revisitation of the Lennox Head site, two years after initial sampling, we were unable to find any S. pinnatifolius individuals. Given the low levels of hybrid seed set, it is most likely that S. madagascariensis achieved dominance via other competitive advantages such as longer flowering time, production of more seeds and greater survival rates (Radford & Cousens, 2000). Both S. madagascariensis and S. pinnatifolius are considered annuals or short lived perennials (Radford & Cousens, 2000), stochastic recruitment failure in annual S. pinnatifolius combined with perennial behaviour in some S. madagascariensis plants could also provide a plausible explanation for the lack of S. pinnatifolius at the study site two years after collection.

Conclusion

Despite limited obvious pre-zygotic isolating barriers restricting hybridisation between the native S. pinnatifolius var. pinnatifolius (dune ecotype) and invasive S. madagascariensis in coastal areas of eastern Australia, we did not find any evidence of adult hybrid plants at two shared sites surveyed in 2007 and analysed with a combination of AFLPs and microsatellites. Hybrid seeds from both S. pinnatifolius and S. madagascariensis were identified at very low levels from open pollinated progeny arrays in the field. Based on these investigations we conclude that introgression of adaptive genes from S. pinnatifolius var. pinnatifolius (dune ecotype) is unlikely to have played a significant role in the success of S. madagascariensis invasions in Australia.

Supplemental Information

Data S1 Spreadsheet containing raw AFLP data used in the study

Click here for additional data file.

Figure S1 STRUCTURE runs for the different AFLP datasets produced with varying error rates

Only allopatric populations of Senecio madagascariensis and Senecio pinnatifolius were used to avoid the confounding effect that detection of hybrids might have on the output. S.madagascariensis is shown on the left side of the plots, S. pinnatifolius ‘dune variant’ on the right. Results are shown for K = 2 (equating to two species) and K = 3 (number of different plates the samples were run on). The dotted lines represent the plate boundaries. The final dataset chosen is shown in (g) where both species are clearly define and at K = 3 there are no obvious plate effects. (a) 0% error, 33 loci; (b) 1% error, 56 loci; (c) 2% error, 79 loci; (d) 3% error, 96 loci; (e) 4% error, 112 loci; (f) 5% error, 128 loci; (g) 6% error, 141 loci; (h) 7% error, 154 loci; (i) 8% error, 165 loci; (j) 9% error, 175 loci; (k) 10% error, 184 loci; (l) 17% error, 233 loci.

Click here for additional data file.

Figure S2 Frequency histogram of locus specific error rates

Frequency histogram of locus specific error rates in the final AFLP dataset with an overall mean error rate of 6% across 141 loci.

Click here for additional data file.

We thank Dr. Ana Pavasovic for her support with field work and Dr. Evelyn White for her communications regarding potential hybrids in the field. We also thank Dr. Gregory Owens and three anonymous reviewers whose constructive reviews greatly improved the quality of the manuscript.

Additional Information and Declarations

Competing Interests

Author Contributions

Data Availability

Peter J. Prentis is an Academic Editor for PeerJ.

Eleanor E. Dormontt conceived and designed the experiments, performed the experiments, analyzed the data, wrote the paper, prepared figures and/or tables.

Peter J. Prentis and Michael G. Gardner conceived and designed the experiments, performed the experiments, analyzed the data, contributed reagents/materials/analysis tools, wrote the paper, reviewed drafts of the paper.

Andrew J. Lowe conceived and designed the experiments, contributed reagents/materials/analysis tools, wrote the paper, reviewed drafts of the paper.

The following information was supplied regarding data availability:

The raw data has been supplied as a Supplementary File.

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
