# Peer review of "Occasional hybridization between a native and invasive Senecio species in Australia is unlikely to contribute to invasive success"

_PeerJ, doi:10.7717/peerj.3630_

## Round 0.1 · original submission · Minor Revisions

The reviewers have made a number of suggestions for revisions. I agree with Reviewer #2 that the findings and methods of Prentis et al 2007 should be more explicitly compared and contrasted with your current study.

I have a few additional minor comments as follows:

L 80 “In previous work…”
L 303 “Fourteen of these hybrid seeds were from S. madagascariensis mothers” shouldn’t this say “..were from a S. madagascariensis mother”? It may then be logical to specify “three from two S. pinnatifolius mothers” and remove the sentence on line 306.
L 328 Identified by what means, DNA or morphology?
L 345 Can anything be inferred about this from Prentis et al 2007?
L 349-350 What where those findings?
L 359 What leads you to suggest this? Please cite reasoning based on this study or Prentis et al 2007.
L398 Doesn’t your finding of so few hybrid seeds in your sampling argue against the likelihood of demographic swamping?
L 413 Delete “preliminary”

·

Basic reporting

The paper is well written, clear and properly referenced. It was an easy read.

Experimental design

The methods were exceptionally detailed. Although the methodology is slightly outdated, it answers the research question sufficiently, and was rigorously vetted.

Validity of the findings

The findings are robust, believable and follow directly from the experimental results.

Additional comments

-Line 63: Missing a space in the brackets.
-Line 118: This citation does not appear in the references.
-Line 349: What are the findings of Scott and White? It is referenced in the introduction, but it would be good to refresh the reader’s memory.
-Line 398: The authors don’t take a firm stance on demographic swamping or competitive advantage. It seems like the data are more indicative of competitive advantage given that the hybridization rate for pinnatifolius mothers was quite low and replacement occurred within 2 years. The authors should elaborate on why it is uncertain or suggest which option is more supported by their data.
-Figure 2: Species names need italicizing.
-Figure 3: This could be structured better. The darker fill is difficult to distinguish from the black outline. Consequently, in 3b it looks like there are off-type individuals at first glance which are actually just dividing lines. I also suggest the authors explicitly label the seedlings versus adult samples instead of just ordering them because currently I can’t tell where the seedling samples begin. It would also be helpful to visually specify which samples were designated as hybrids in this figure.

Reviewer 2 ·

Basic reporting

No comment

Experimental design

Knowledge gap is a problem given that similar study was done 10 years ago.

Validity of the findings

However, these questions were asked and answered in a previous study by some of the same authors (Prentis and Lowe) examining another variety of Senecio pinnatifolius “Tableland” (Prentis et al 2007). And therein arises my concerns with the present paper; what distinguishes this study from the previous one?

Although meaningful replication of results is acceptable to this journal, I think the authors should emphasize just where and why their results are different from the previous study

Additional comments

Review of Dormontt et al “Occasional hybridization between a native and invasive species pair unlikely to contribute to invasive success”

This study examines the frequency of hybrids, both adults and progeny, between native Senecio pinnatifolius var pinnatifolius and exotic S. madagascariensis in Australia to answer the question as to whether hybrid seed set occurs and whether adult hybrids are present in nature. This well written study was purely based on genetics and the techniques, results, and interpretations are sound.

However, these questions were asked and answered in a previous study by some of the same authors (Prentis and Lowe) examining another variety of Senecio pinnatifolius “Tableland” (Prentis et al 2007). And therein arises my concerns with the present paper; what distinguishes this study from the previous one? The previous study used reciprocal hand pollinations to look at intra- and interspecific seed set, examined germination, and used molecular methods to look at genetic structure and to identify hybrids, both arising in nature and from cross pollinations; they found bi-directional hybridization occurred but no adults were found in nature. The present study used molecular methods to identify hybrid adults and seedling progeny in sympatric populations after first developing markers from allopatric populations. What the present study finds is bi-directional hybridization occurred but no adults were found in nature.

Although meaningful replication of results is acceptable to this journal, I think the authors should emphasize just where and why their results are different from the previous study; the why will require the use of non-genetic tools in additional research. It could be the authors have fallen prey to Maslow's hammer, "if the only tool you have is a hammer, you treat everything as if it were a nail," in that suggestions for future research include more genetics (Next-Gen sequencing). However, they do suggest other non-genetic avenues that could answer the basic puzzle of the population dynamics of the two species and their hybrids. This approach may yield results that aid in forestalling the spread of the exotic (if possible!) or encourage the spread of the native.

The three interesting results they found that distinguish their study from the earlier one are 1, that a single S. m mother produced ALL the S. m mothered hybrid progeny at LH, 2, hybrid formation by the native was rare and 3, two years after the study, the native species had vanished from one of their study sites. These suggest future approaches to me, to wit:
1. The ability to set seed can vary widely among individual plants. I have found in my work with wild sunflowers (and grasses) that many florets in an inflorescence contain no embryos; this can be assessed by lightly finger pressing the florets after deconstructing the inflorescence. I have used single inflorescences as replicates to estimate seed set per individual and this varies a LOT – some individuals are far better “mothers” than others; some individuals NEVER set seed. When I test germination I only use florets with an embryo. Each individual plant thus has a measure of seed set and seed germination to generate an evaluation of viable seed set. I have found that just a few plants can contribute the majority of viable seeds in a population. This may have relevance to your system as it appears that a single S. m plant is a very good mother to hybrids while the rest set NO hybrid seed.
2. The previous study suggested that the exotic had superior siring ability than the native and even small numbers of S. m could reproductively dominate in a mixed population leading to the extinction of the native. This was clearly NOT the case here. Why not? A, pre-zygotic barriers were stronger – test using reciprocal crosses; b, pollinator behavior influenced interspecific crossing – observe pollinators in the field as they suggest.
3. As #2 was not happening at their site, why did the native vanish from the mixed LH population? I have seen S. m listed as an annual, biennial, and short-lived perennial. I have also seen the native listed as annual or perennial. This basic life history is not mentioned in the paper and could provide clues. If the exotic was behaving as a perennial and the native as an annual, then recruitment failure for two years, perhaps due to drought, would explain the native's absence. What are the regeneration niches (sensu Grubbs) of the two species? The fitness of pure and hybrid individuals in the common garden experiment they suggest may explain not only the lack of hybrid plants, but the relative fitness of each species.

Other issues:
1. Methods – Line 146, “mature seed heads were collected from sampled plants”. Then, line 152, “In the laboratory, mature achenes were detached from their pappus and the seed coat nicked with a scalpel” to promote germination, resulting in “Up to twenty seedlings per parent plant” for genetic analysis. Was the collection of seeds from each parent a combination of multiple seed heads or was a single seed head used for the seed collection? I ask because if the seeds from a single capitulum were genotyped, then this could represent a single visit by a pollinator. However, if the seeds from several capitulums were pooled, then this would suggest continued and repeated interspecific pollinations by insects.
2. Did germination vary between individuals?
3. Fig. 1 contains a lot of spatial and genetic information. However, the proportion of hybrid seed within a genotyped plant is very difficult to discern – I had to enlarge to 200% to see that. When I did I saw that:
a. a single individual at Lennox Head (LH) contained 2/3 hybrid (=13) and 1/3 S.m seedlings out of 19 seedlings genotyped. Naturally, as a S.p plant cannot give rise to pure S.m progeny this individual MUST be a S. m individual embedded in a S. p group. However, in Fig. 2b the maternity of these 13 hybrid seedlings at LH is attributed to S.p (denoted by Xs) instead of S. m (denoted by +s).
b. 2 adults of S. p gave rise to 3 hybrids, yet in Fig. 2b the maternity of these hybrids is given as pluses denoting S. m maternity instead of S. p maternity.
c. However, in the text, lines 302-306, the correct percentages and maternity are given.
4. Why are there no pure species seedlings from Lennox Head in Fig. 2b?
5. Table 1 needs a legend – what do nsuba, nsubs, and s (range) mean?
6. Typos: Line 82 “predicted” instead of predicting
Line 358 C. Jacea should be C. jacea

Prentis PJ, White EM, Radford IJ, Lowe AJ, and Clarke AR. 2007. Can hybridization cause local extinction: a case for demographic swamping of the Australian native Senecio pinnatifolius by the invasive Senecio madagascariensis? New Phytologist 176:902–912

---

## Round 0.2 · Minor Revisions

The authors have addressed the reviewers comments to the extent possible. As a final comment, I think the authors should carefully consider the wording of the title.

1) It basically is a complete sentence that is grammatically incorrect because the verb "is" is missing in front of "unlikely". Why not add the verb?
2) Is the noun "pair" needed?
3) The title isn't very informative. Might you add "Sencio" somewhere, or "plant"? I understand that the authors might want to deliberately use a vague title in the hope of attracting more readers - if they feel that is they want to go then ok to keep it vague.

---

## Round 0.3 · accepted · Accept

The revised title is clear and informative.